# Design of Optical Tunnel Switching Networks for Big Data Applications

**Yuh-Jiuh Cheng** [1,*] **, Bor-Tauo Chen** [1] **, Cheng-Ping Wu** [1] **and Yu-Yun Lee** [2]

[1] Broadband Networks Laboratory, Telecommunication Laboratories, Chunghwa Telecom Co., Ltd., Taoyuan City 32601, Taiwan; ptchen@cht.com.tw (B.-T.C.); cpwu@cht.com.tw (C.-P.W.)

[2] Accton Technology Corporation, Hsinchu Science Park, Hsinchu 30077, Taiwan; leeyuyun@accton.com

[*] Correspondence: yjcheng@cht.com.tw; Tel.: +886-03-4244158

**Abstract:** In this paper, we proposed large-scale optical tunnel switching networks based on the Torus topology network with WSS (Wavelength Selective Switch) for future big data applications. All nodes of the large-scale optical tunnel switching networks use WSS switch modules, and the communications between nodes use multiple λs (wavelengths), where a tunnel is established with a wavelength which can be reused. The widely used MEMS (Micro-Electro-Mechanical Systems) and LCoS (Liquid Crystal on Silicon) technologies are all millisecond-level switching speeds, so the frame size of the optical frame switch is very large, and this will reduce switching performance. Therefore, they are only suitable for optical tunnel switching networks design, but are not suitable for optical frame switch design. This multi-plane Torus topology network architecture not only increases network throughput, but also has fault tolerance to increase network reliability. When the traffic is changed, the number of tunnels between nodes can be scheduled in time to balance the load traffic and avoid traffic loss. Therefore, it can not only schedule the number of tunnels in time to balance the load traffic, in order to avoid traffic loss, but also because the channel is fixedly established, this does not generate any buffer delay, and this because of the transmission using optical transmission unlimited speed, so it is a good choice for future big data applications that require high speed, high bandwidth and low latency.

**Keywords:** wavelength selective switch; torus topology network; optical tunnel switching network

## 1. Introduction

Currently, the companies focus on seven Artificial Intelligence (AI) applications, including chat bots, process advancement and fraud detection, face recognition, market segmentation, sentiment analysis and call center assistants. The Xiaomi AI speaker has six radio microphones for all-round radio and listening to voice commands. In addition to voice-activated music or listening to audio books, you can also ask for exchange rates, weather, etc., or ask it to calculate math problems. These applications require high speed, high bandwidth, low latency network systems.

Due to the success of streaming video and music service modes, as well as the increase in network bandwidth, streaming game services will be launched in the future. In the past, manufacturers failed to promote streaming games, partly because of the "delay" problem. So the spread of data centers and the utility of the new generation of mobile network 5G will help to meet these challenges.

Unmanned aerial vehicles (UAVs) are very mature in product development, but they have not been widely applied so far, mainly because of the lack of a systematic management platform, resulting in the inability to effectively supervise a large number of drones.

The accuracy of traffic jam forecasting on specific road sections has now reached 90%, which allows driving to effectively avoid traffic jams, but sometimes it may cause alternative road traffic jams,

and returning home is still a time-consuming shortcoming of the herd effect. All-round applications, such as smart retail, smart access control and smart city, will create the most intelligent and convenient artificial intelligence Internet of Things (AIoT) life experience. FaceMe's AI face recognition engine has a correct recognition rate of 98.41% in its ultra-high precision mode. FaceMe has the advantage of edge computing integration. The AIoT device can transfer data processing from the cloud to the edge computing, without relying on the cloud server, greatly improving the speed of face recognition.

The rapid growth of the Internet, mobile networks and cloud services, such as 3D games, VR (Virtual Reality), AR (Augmented Reality) and cloud services, will increase the bandwidth requirements, and network latency issues are also more demanding. In addition, there is a smart controller that can control smart appliances and devices, including electric cookers, sweeping robots, smart desk lamps, air purifiers, smart sockets, artificial intelligence Internet of Things (AIoT), and so on. These new applications which yield unprecedented volumes of exchanged data have been driving the scale of data centers and the diversity of services. Current data center networks are unable to meet demand due to their insufficient performance, such as poor flexibility, high latency, lack of scalability and serious power consumption. In recent years, various optical switching systems have been developed and demonstrated, making them the best choice for next-generation data center and data center networking technologies [1–10]. Test results show that compared with the electric spine-leaf data center network, OPTUNS achieves 82.6% power savings under high traffic load (locality), and significantly reduces the average (p99) delay [11]. The Torus topology data center network, using hybrid OPS/OCS (Optical Packet Switch/ Optical Circuit Switch) technology, proposes an energy-efficient, low-latency, data center internal network [8].

The optical frame switch has been discussed [12]. After the simulation, the $10 \times 10$ Torus optical frame switch requires about 24 $\lambda$s per link. One node has four links, so the total of 96 $\lambda$s is required to reach a throughput of nearly 100%. However, optical frame switches require faster optical switching times. For example, under a 100 Gbps link, if the frame size is 1.25 Kbytes, optical switching components with a switching time of 100 nanoseconds or less are required, such as the Semiconductor Optical Amplifier (SOA) optical switching components with switch time to 1 nanosecond. The widely used MEMS and LCoS technologies are all millisecond-level switching speeds, so the frame size of the optical frame switch is very large. For example, under a 100 Gbps link, the frame size should be at least 12.5 Mbytes, which will cause a large delay in the optical frame switch, thus affecting the performance of the optical frame switch.

Therefore, an optical tunnel switch is proposed in this paper. This technology mainly uses Dense Wavelength Division Multiplexer (DWDM) technology to establish many tunnels with multiple wavelengths to provide point-to-point connections. Therefore, optical tunnel switches require many wavelengths to make point-to-point, non-blocking connections.

The proposed architecture of the optical tunnel switching network in this paper is to establish a wavelength tunnel between nodes. This wavelength tunnel is not used by different nodes in time sharing, but the tunnel can be shared by multiple nodes. When the traffic volume of the nodes which are monitored exceeds the traffic volume of the wavelength tunnel, the wavelengths of other nodes with less traffic volume can be scheduled for use. Therefore, we calculate the traffic volume between nodes, and the average available traffic of the server. When the average available traffic is exceeded, wavelength scheduling is performed. For example, if the two plan optical tunnel switching network is adopted, there are two direct tunnels between nodes. The two direct tunnels can be used to communicate between nodes, and they are responsibility to establish some relay tunnels for other nodes, because tunnels must be established between all nodes in our proposed architecture. If the optical tunnel switching network has 49 nodes, then $49 \times 48$ tunnels are required to be established between all nodes. Therefore, if a tunnel traffic is exceeded, we can assign other tunnels with less traffic to support this busy tunnel. All of the operations will be handled by a smart scheduler which may be a Software Defined Networking (SDN) controller, including an algorithm or methodology. The optical frame switching network allows each wavelength channel to be used by different nodes at

different times. Therefore, a scheduling algorithm must be used to control the usage of every node and simulate its performance, as shown in [12]. Its performance is related to the frame size. Because of the smaller frame with the better performance, therefore, high-speed switching elements, such as SOA switching elements, are required. The optical tunnel switching network is only used for wavelength tunnel scheduling, and does not require high-speed switching elements. Therefore, the performance of the optical tunnel switching network is necessary that the average throughput of traffic reaches to 100% at least. However, the performance of the optical frame switching network ensures that average throughput is less than 100%, because the aggregation algorithm is adopted. This is because, when much traffic is sent to the same node, some traffic will be dropped. In the optical tunnel switching network, when a lot of traffic gets sent to the same node, we can schedule the wavelength tunnel to avoid traffic loss. The advantages of the optical tunnel switching network is that it does not require high-speed switching elements and flexible channel scheduling to avoid data loss when the traffic volume is changed.

The brief descriptions of other sections are presented as follows. In Section 2, we describe the operations of a multi-plane Torus topology optical tunnel switching network. In Section 3, a two-plane Torus topology optical tunnel switching network with 25 nodes is presented, and its performance is discussed. The three-plane Torus topology optical tunnel switching network with 49 nodes is presented, and its performance is also discussed in Section 4. In Section 5, performance evaluation, traffic management and applications of optical tunnel switching networks are discussed. Finally, some conclusions are made in Section 6.

## 2. Torus Optical Tunnel Switch Network

The structure of the wavelength selective switch based on LCoS technology is shown in Figure 1 (Provided by Accton Technology Corp.). The SDN technology is used to allocate a wavelength of the input port to the specific output port. For example, $\lambda_1$, $\lambda_7$, $\lambda_{11}$ and $\lambda_{20}$ of input port 1 are assigned to transfer to output port 3, and so on.

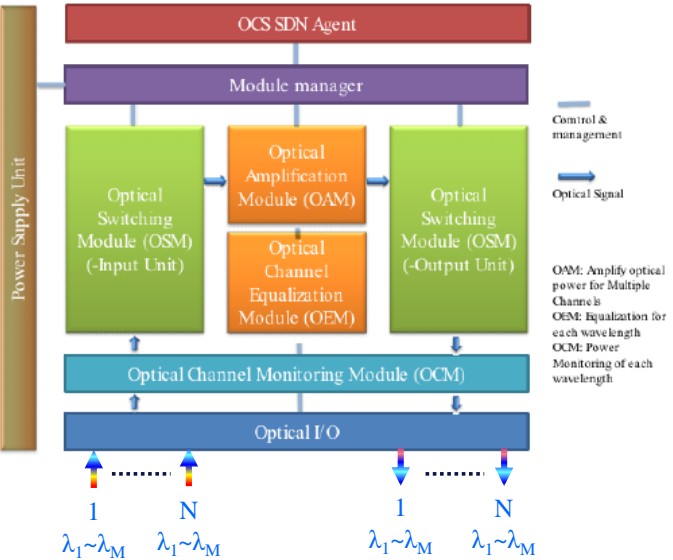

**Figure 1.** Architecture of the wavelength selective switch.

Figure 2 shows a multi-plane Torus topology optical switching network. The X-axis has X nodes and the Y-axis has Y nodes, so there is a total of X × Y nodes. In order to improve the throughput, Z planes are used.

X-axis has X nodes that need X × (X-1) different wavelengths (λ) if we need to establish full match tunnels. Similarly, Y-axis needs Y × (Y-1) different wavelengths if we need to establish full match tunnels. Therefore, the total number of wavelengths named W is defined:

$$W = X \times (X-1) + Y \times (Y-1), \tag{1}$$

Let Y = X

$$W = X \times (X-1) + X \times (X-1) = 2X^2 - 2X, \tag{2}$$

After calculation, we can obtain

$$X = \frac{1 + \sqrt{2 \times W + 1}}{2}, \tag{3}$$

Therefore, if W = 48, X = 5.42 takes its integer, and the maximum number of nodes is 25 (5 × 5). Similarly, if W = 96, X = 7.3 takes its integer, and the maximum number of nodes is 49 (7 × 7). If we use these wavelengths, we can create direct tunnels. For example, if X = Y = 5, the direct tunnels of each node to the other nodes are only 8 (4 + 4), but the tunnels of each node to the other nodes need 24, so the other 16 tunnels must be transferred through the relay nodes. The total number of tunnels needs 600 (24 × 25). Similarly, if X = Y = 7, the direct tunnels of each node to other nodes is only 12 (6 + 6), but the tunnels of each node to the other nodes need 48, so the other 36 tunnels must be transferred through the relay nodes. The total number of tunnels requires 2352 (48 × 49). These relay tunnels occupy the bandwidth of the direct tunnel, so multiple planes must be used to increase the network throughput. How many planes are needed depend on the number of nodes, and the required throughput that will be discussed in Sections 3 and 4.

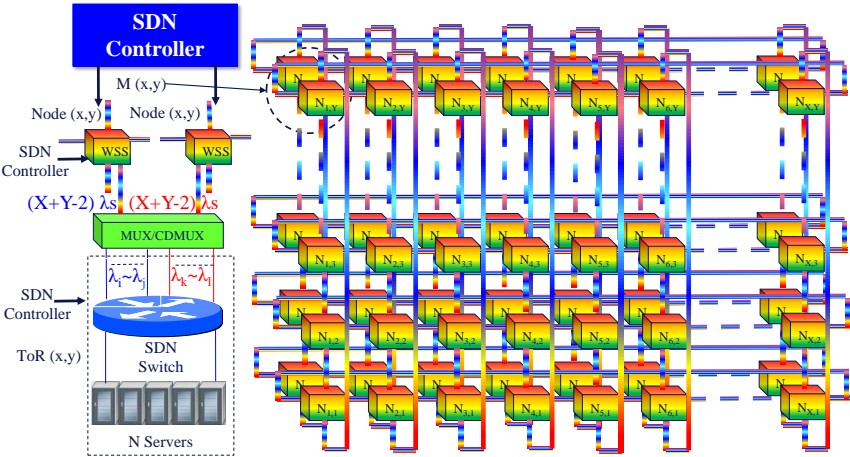

**Figure 2.** Multi-plane Torus topology optical tunnel switching network.

Each module is composed of N servers, Z MUXs (Multiplexer), Z CDMUXs (Cyclic De-Multiplexer), an SDN switch and Z nodes with a node in each plane as shown in Figure 3, where each node is a 5 × 5 WSS. This multi-plane architecture not only increases the throughput of the network, but also increases the stability of the network, and has fault tolerance capability. A module can be connected to N servers. These servers collect data and sort through an SDN switch to aggregate the data to the same module and then transfer it to the multiplexer. The (X + Y − 2) wavelengths are grouped into a DWDM signal and sent to a 5 × 5 wavelength selective switch (5 × 5 WSS). Each node has a 5 × 5 WSS to set the different wavelengths to send to different nodes to establish a tunnel. Different wavelengths in the DWDM signal receiving from different nodes are decomposed into the SDN switch via a CDMUX, and then sent to their corresponding servers. Other planes have the same function to increase network throughput, and when a plane fails, traffic can be routed to another plane to increase network reliability and stability.

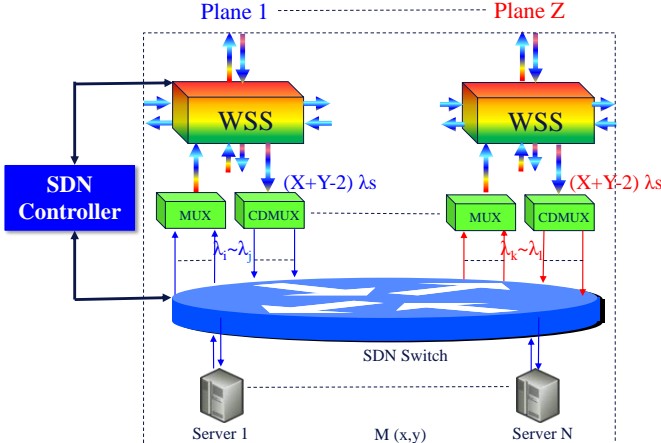

**Figure 3.** A module of multi-plane Torus topology optical tunnel switching network.

## 3. Torus Optical Tunnel Switch Network with 25 Nodes

Figure 4 shows a two-plane Torus topology optical switching network with 25 nodes. Each module is composed of 16 servers, 2 MUXs, 2 CDMUXs, an SDN switch and 2 nodes, with a node in each plane, as shown in Figure 5, where each node is a $5 \times 5$ WSS.

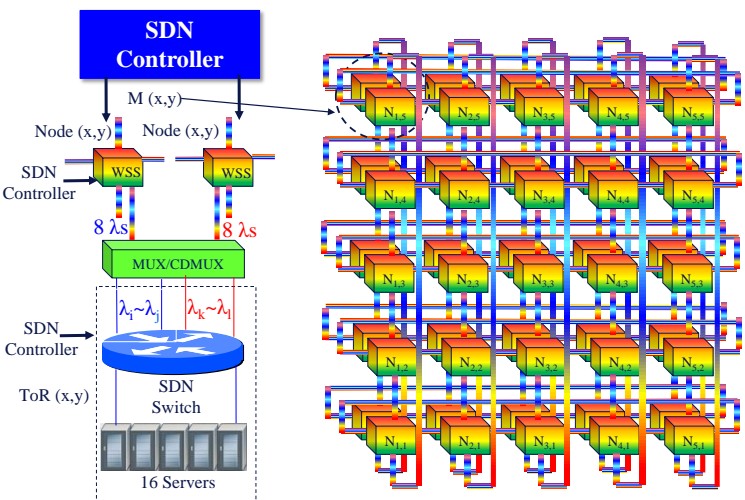

**Figure 4.** Two-plane Torus topology optical tunnel switching network with 25 nodes.

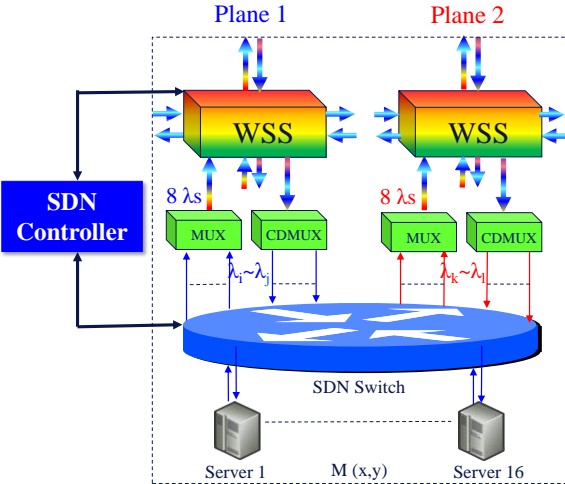

**Figure 5.** A module of two-plane Torus topology optical tunnel switching network.

The X-axis and Y-axis wavelength assignments of each node are shown in Table 1. A total of 40 wavelengths are shared and reused with different planes. Figure 6 shows the first plane tunnel creation of a two-plane Torus topology optical tunnel switching network with 25 nodes. Suppose node N (1, 5) wants to build a tunnel to node N (4, 5), because the two nodes are in the same X-axis loop, and therefore $\lambda_3$ (refer to the blue word in Table 1) can establish a direct tunnel. But node N (1, 5) needs to build a tunnel to node N (5, 3), because they are not in the same X-axis loop, therefore it is necessary to establish a relay tunnel; that is, first establish a direct tunnel to node N (5, 5) with $\lambda_4$ (refer to the red word in Table 1), and transfer to the SDN switch and then transfer to node N (5, 3) with $\lambda_{38}$ (refer to the red word in Table 1). N (1, 5) also needs to build a direct tunnel to node N (5, 5), and therefore, the tunnel from the node N (1, 5) to the node N (5, 5) is shared with the previous relay tunnel. That is to say, the traffic volume of the two tunnels will be integrated, so the average traffic volume cannot exceed 50%. Of course, the SDN switch at node N (5, 5) must separate the two nodes' traffic according to different IP addresses into the correct servers of the two nodes. That is, the traffic to the node N (5, 5) is left, and the traffic to the node N (5, 3) is transferred out via the SDN switch.

**Table 1.** X-axis and Y-axis wavelength assignment of each node.

| Node | X-Axis Wavelength | | | | Y-Axis Wavelength | | | |
|---|---|---|---|---|---|---|---|---|
| N (1, 5) | $\lambda_1$ | $\lambda_2$ | $\lambda_3$ | $\lambda_4$ | $\lambda_5$ | $\lambda_6$ | $\lambda_7$ | $\lambda_8$ |
| N (2, 5) | $\lambda_9$ | $\lambda_{10}$ | $\lambda_{11}$ | $\lambda_{12}$ | $\lambda_{13}$ | $\lambda_{14}$ | $\lambda_{15}$ | $\lambda_{16}$ |
| N ( 3, 5) | $\lambda_{17}$ | $\lambda_{18}$ | $\lambda_{19}$ | $\lambda_{20}$ | $\lambda_{21}$ | $\lambda_{22}$ | $\lambda_{23}$ | $\lambda_{24}$ |
| N (4, 5) | $\lambda_{25}$ | $\lambda_{26}$ | $\lambda_{27}$ | $\lambda_{28}$ | $\lambda_{29}$ | $\lambda_{30}$ | $\lambda_{31}$ | $\lambda_{32}$ |
| N (5, 5) | $\lambda_{33}$ | $\lambda_{34}$ | $\lambda_{35}$ | $\lambda_{36}$ | $\lambda_{37}$ | $\lambda_{38}$ | $\lambda_{39}$ | $\lambda_{40}$ |
| N (1, 4) | $\lambda_9$ | $\lambda_{10}$ | $\lambda_{11}$ | $\lambda_{12}$ | $\lambda_{13}$ | $\lambda_{14}$ | $\lambda_{15}$ | $\lambda_{16}$ |
| N (2, 4) | $\lambda_{17}$ | $\lambda_{18}$ | $\lambda_{19}$ | $\lambda_{20}$ | $\lambda_{21}$ | $\lambda_{22}$ | $\lambda_{23}$ | $\lambda_{24}$ |
| N (3, 4) | $\lambda_{25}$ | $\lambda_{26}$ | $\lambda_{27}$ | $\lambda_{28}$ | $\lambda_{29}$ | $\lambda_{30}$ | $\lambda_{31}$ | $\lambda_{32}$ |
| N (4, 4) | $\lambda_{33}$ | $\lambda_{34}$ | $\lambda_{35}$ | $\lambda_{36}$ | $\lambda_{37}$ | $\lambda_{38}$ | $\lambda_{39}$ | $\lambda_{40}$ |
| N (5, 4) | $\lambda_1$ | $\lambda_2$ | $\lambda_3$ | $\lambda_4$ | $\lambda_5$ | $\lambda_6$ | $\lambda_7$ | $\lambda_8$ |
| N (1, 3) | $\lambda_{17}$ | $\lambda_{18}$ | $\lambda_{19}$ | $\lambda_{20}$ | $\lambda_{21}$ | $\lambda_{22}$ | $\lambda_{23}$ | $\lambda_{24}$ |
| N (2, 3) | $\lambda_{25}$ | $\lambda_{26}$ | $\lambda_{27}$ | $\lambda_{28}$ | $\lambda_{29}$ | $\lambda_{30}$ | $\lambda_{31}$ | $\lambda_{32}$ |
| N (3, 3) | $\lambda_{33}$ | $\lambda_{34}$ | $\lambda_{35}$ | $\lambda_{36}$ | $\lambda_{37}$ | $\lambda_{38}$ | $\lambda_{39}$ | $\lambda_{40}$ |
| N (4, 3) | $\lambda_1$ | $\lambda_2$ | $\lambda_3$ | $\lambda_4$ | $\lambda_5$ | $\lambda_6$ | $\lambda_7$ | $\lambda_8$ |
| N (5, 3) | $\lambda_9$ | $\lambda_{10}$ | $\lambda_{11}$ | $\lambda_{12}$ | $\lambda_{13}$ | $\lambda_{14}$ | $\lambda_{15}$ | $\lambda_{16}$ |
| N (1, 2) | $\lambda_{25}$ | $\lambda_{26}$ | $\lambda_{27}$ | $\lambda_{28}$ | $\lambda_{29}$ | $\lambda_{30}$ | $\lambda_{31}$ | $\lambda_{32}$ |
| N (2, 2) | $\lambda_{33}$ | $\lambda_{34}$ | $\lambda_{35}$ | $\lambda_{36}$ | $\lambda_{37}$ | $\lambda_{38}$ | $\lambda_{39}$ | $\lambda_{40}$ |
| N (3, 2) | $\lambda_1$ | $\lambda_2$ | $\lambda_3$ | $\lambda_4$ | $\lambda_5$ | $\lambda_6$ | $\lambda_7$ | $\lambda_8$ |
| N (4, 2) | $\lambda_9$ | $\lambda_{10}$ | $\lambda_{11}$ | $\lambda_{12}$ | $\lambda_{13}$ | $\lambda_{14}$ | $\lambda_{15}$ | $\lambda_{16}$ |
| N (5, 2) | $\lambda_{17}$ | $\lambda_{18}$ | $\lambda_{19}$ | $\lambda_{20}$ | $\lambda_{21}$ | $\lambda_{22}$ | $\lambda_{23}$ | $\lambda_{24}$ |
| N (1, 1) | $\lambda_{33}$ | $\lambda_{34}$ | $\lambda_{35}$ | $\lambda_{36}$ | $\lambda_{37}$ | $\lambda_{38}$ | $\lambda_{39}$ | $\lambda_{40}$ |
| N (2, 1) | $\lambda_1$ | $\lambda_2$ | $\lambda_3$ | $\lambda_4$ | $\lambda_5$ | $\lambda_6$ | $\lambda_7$ | $\lambda_8$ |
| N (3, 1) | $\lambda_9$ | $\lambda_{10}$ | $\lambda_{11}$ | $\lambda_{12}$ | $\lambda_{13}$ | $\lambda_{14}$ | $\lambda_{15}$ | $\lambda_{16}$ |
| N (4, 1) | $\lambda_{17}$ | $\lambda_{18}$ | $\lambda_{19}$ | $\lambda_{20}$ | $\lambda_{21}$ | $\lambda_{22}$ | $\lambda_{23}$ | $\lambda_{24}$ |
| N (5, 1) | $\lambda_{25}$ | $\lambda_{26}$ | $\lambda_{27}$ | $\lambda_{28}$ | $\lambda_{29}$ | $\lambda_{30}$ | $\lambda_{31}$ | $\lambda_{32}$ |

Note: The $\lambda_3$ is used to establish a direct tunnel between node N (1, 5) and node N (4, 5). The $\lambda_4$ and $\lambda_{38}$ are used to establish a relay tunnel between node N (1, 5) and node N (5, 3) via node N (5, 5).

However, if each direct tunnel is responsible for transferring a relay tunnel node, there are still 8 nodes that cannot receive traffic from node N (1, 5), so if each direct tunnel is responsible for transferring two relay tunnel nodes, then node N (1, 5) can transmit traffic to any other 24 nodes. However, sharing a direct tunnel between the three nodes reduces the average throughput of traffic to 33% approximately. Therefore, it is necessary to add a plane Torus topology optical tunnel switching network to increase the throughput of traffic. If two planes of the Torus topology optical switching network are used, each plane is responsible for one relay tunnel node, which is equal to two direct tunnels to three nodes, so the throughput of traffic can reach an average of about 66%. Moreover, one module shares two nodes, and one module has only 16 servers. If these servers are evenly distributed

to 24 modules, the average throughput of traffic is only about 66%. Therefore, the two-plane Torus topology optical switching network is sufficient to provide 100% of the server's average throughput of traffic. Of course, if the traffic is not evenly distributed, the distribution of the tunnels must be adjusted.

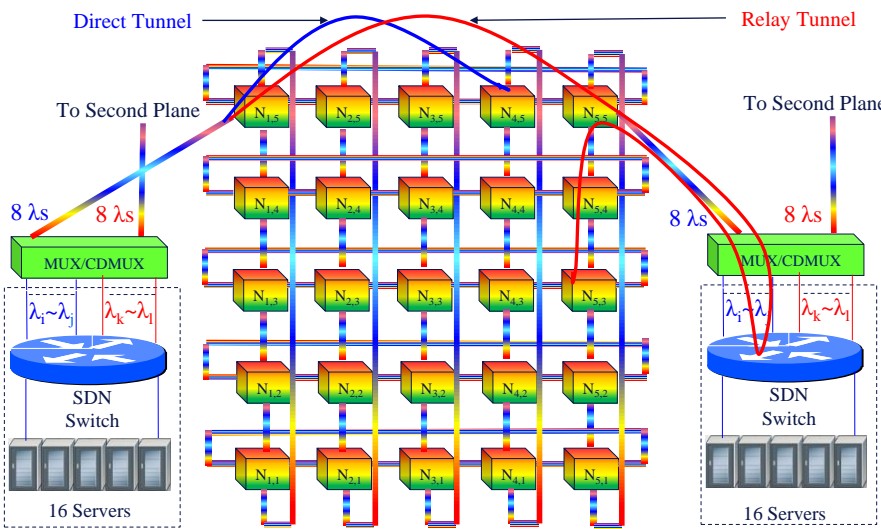

**Figure 6.** Tunnel creation of first plane.

## 4. Torus Optical Tunnel Switch Network with 49 Nodes

To increase the capacity of the network, Figure 7 shows a three-plane Torus topology optical tunnel switching network with 49 nodes. Figure 8 shows the module of the three-plane Torus topology optical tunnel switching network with 49 nodes. Each module is composed of 32 servers, 3 MUXs, 3 CDMUXs, an SDN switch and 3 nodes, with a node in each plane, as shown in Figure 8, where each node is a 5 × 5 WSS. Each module consists of three planes corresponding nodes, so 12 λs can establish 6 direct tunnels for the X-axis and Y-axis, respectively. In addition, there are 36 relay tunnel nodes for each module to be established. Similarly, if each plane is responsible for transferring a relay tunnel node, the three direct tunnels are responsible for the traffic of 4 modules, so the average throughput is 75% (3/4). Each module has 36 servers that average to send to 48 modules with an average throughput of only 75% (36/48). Therefore, the three-plane Torus topology optical switching network is sufficient to provide 100% of the average traffic of the server.

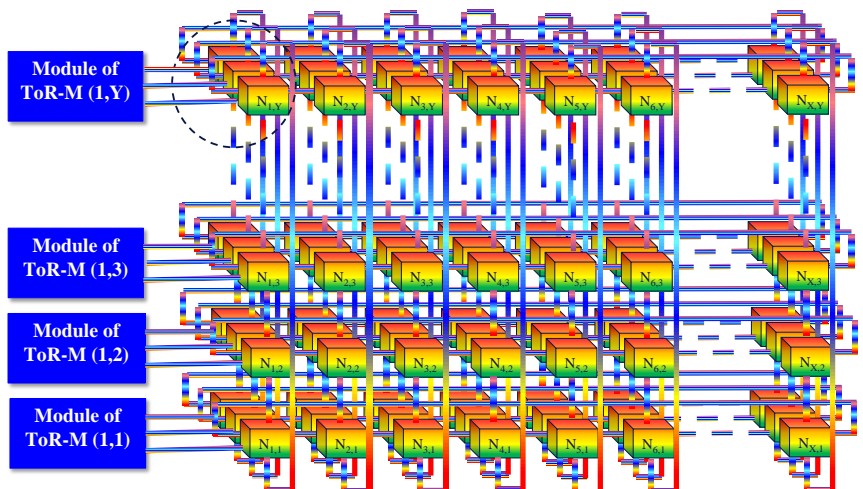

**Figure 7.** Three-plane Torus topology optical tunnel switching network with 49 nodes.

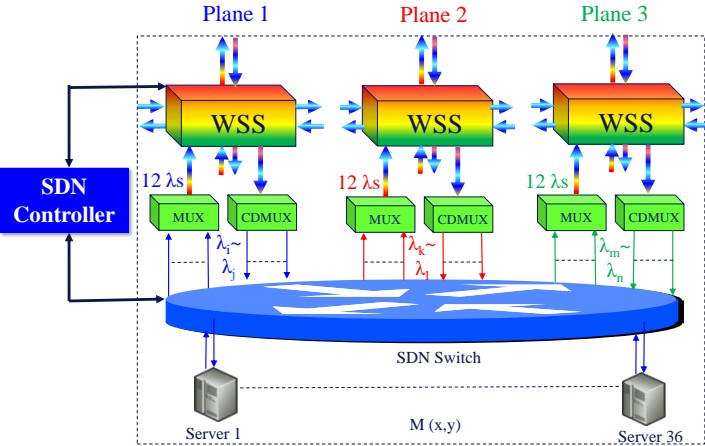

**Figure 8.** A module of Three-plane Torus topology optical tunnel switching network.

Figure 9 shows tunnel creation of a three-plane Torus topology optical tunnel switching network with 49 nodes for the first plane. Let X = Y = 7. Therefore, node N (1, 7) and node N (6, 7) can establish a direct tunnel, and node N (1, 7) and node N (7, 2) need to establish a relay tunnel through node N (7, 7). Similarly, node N (1, 3) and node N (7, 3) can establish a direct tunnel, and node N (1, 3) and node N (6, 1) need to establish a relay tunnel through node N (6, 3). Other nodes can also establish direct tunnels and relay tunnels in sequence.

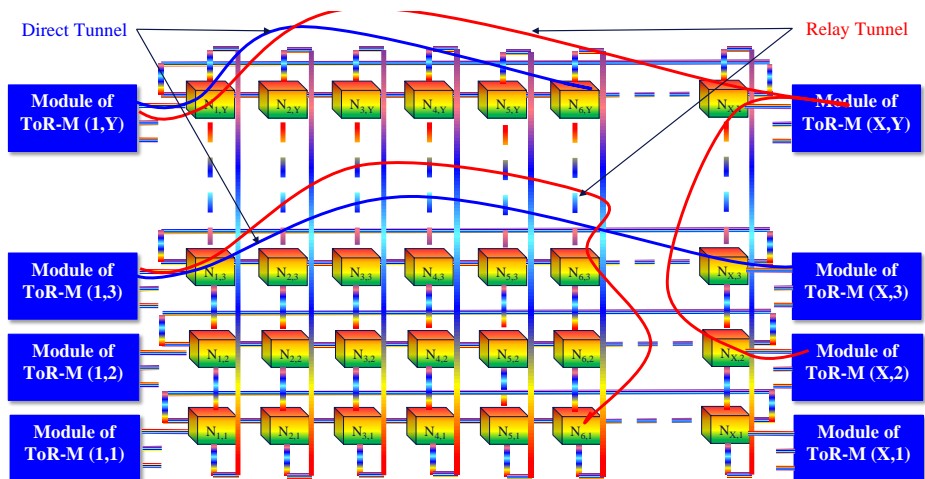

**Figure 9.** Tunnel creation of first plane.

The three-plane Torus topology optical switching network can greatly increase the capacity of the network. This architecture utilizes 84 λs to provide 1764 servers and establish 2352 tunnels. The architecture in [11] utilizes 40 λs to provide 400 servers and establish 600 tunnels. So, while the number of wavelengths has only doubled, the number of servers that it provides can be increased by more than four times. In the future, if the number of wavelengths can be increased, the number of servers provided can be increased.

## 5. Network Performance and Its Applications

### 5.1. Performance Evaluation

For a detailed description of the switch performance, Figure 10 shows the tunnel assignment of a two-plane optical tunnel switching network with 25 nodes. Each node of the 5 × 5 Torus network has 8 direct tunnels and 16 relay tunnels to transfer data to other nodes. According to the uniform distribution of traffic, on average, a direct tunnel allows two relay tunnels (16/8) to share bandwidth,

such as node N (5, 5), node N (5, 4) and node N (5, 3) share a direct tunnel from node N (1, 5) to node N (5, 5), as well as node N (1, 4), node N (2, 4) and node N (4, 4) share another direct tunnel from node N (1, 5) to node N (1, 4). If a plane uses only a relay tunnel to share the bandwidth of the direct tunnel, the network needs dual planes. Therefore, we can connect from node N (1, 5) to all nodes on the network at the same time. These three nodes will share the bandwidth of the direct tunnel, and each node will use an average of two-thirds of the bandwidth of the two planes. One SDN switch per node can use 16 wavelength bandwidths for 16 servers (2 × 8). If the 16 wavelength bandwidths of the 16 servers are transmitted to the other 24 node modules on average, the average traffic is also 2/3 of the wavelength bandwidth (16/24).

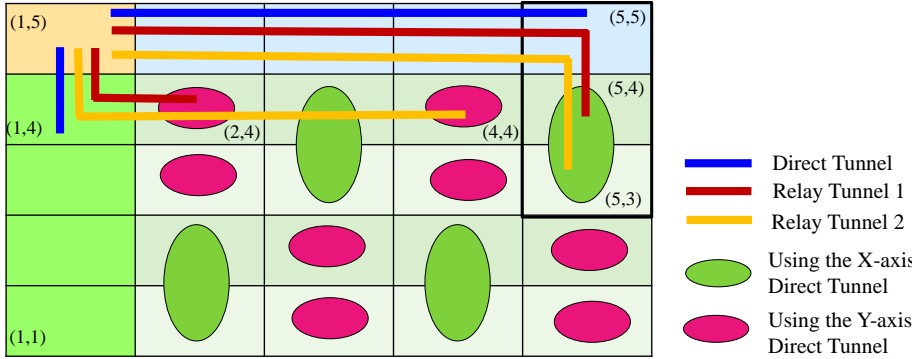

**Figure 10.** Tunnel assignment of a two-plane optical tunnel switching network with 25 nodes.

Figure 11 shows the tunnel assignment of a three-plane optical tunnel switching network with 49 nodes. Each node of the 7 × 7 Torus network has 12 direct tunnels and 36 relay tunnels to transfer data to other nodes. According to the uniform distribution of traffic, on average, a direct tunnel allows three relay tunnels (36/12) to share bandwidth, such that node N (7, 7), node N (7, 6), node N (7, 5) and node N (7, 4) share a direct tunnel, and the tunnel runs from node N (1, 7) to node N (7, 7). If one plane uses only a relay tunnel to share the bandwidth of the direct tunnel, the network requires three planes. Therefore, we can connect from node N (1, 7) to all nodes on the network at the same time. The four nodes will share the bandwidth of the direct tunnel, and each node uses an average of 3/4 wavelength bandwidth across the three planes. One SDN switch per node can use 36 wavelength bandwidth for 36 servers (3 × 12). If these 36 wavelength bandwidths for 36 servers are transmitted to the other 48 node modules on average, the average traffic is also 3/4 of the wavelength bandwidth (36/48).

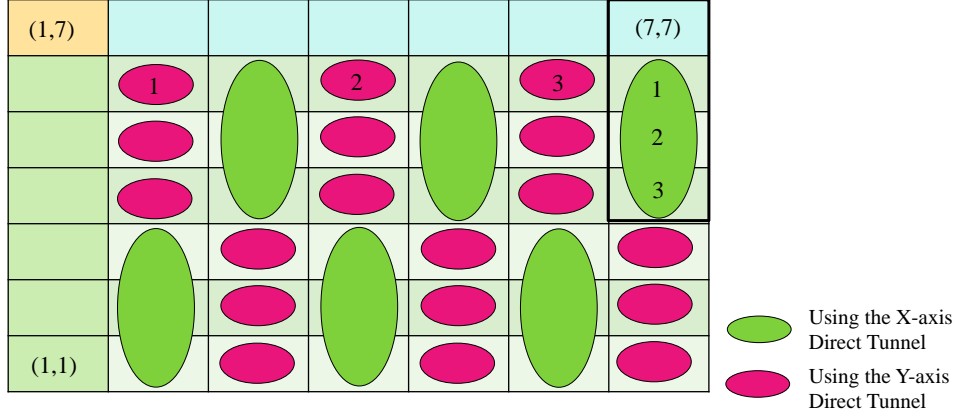

**Figure 11.** Tunnel assignment of a three-plane optical tunnel switching network with 49 nodes.

To investigate the performance of optical tunnel switching networks, we capture traffic from the Internet core network of Chunghwa Telecom Co., Ltd. in Taiwan (a major provider of telecommunications services). According to the data captured from the core network of a traditional

Ethernet switch, the traffic of Ethernet packets is shown in Figure 12, where the X-axis represents various packet sizes. The number of packets of various packet sizes is shown at the bottom of Figure 12, the total number of bytes (TN) of various packet sizes is shown in the middle of Figure 12, and the cumulative total number of bytes (CTN) of various packet sizes is shown at the top of Figure 12. In other words, if the number of packets is equal to N*i* for various packet sizes, where *i* is denoted the packet size. For various packet sizes, the characteristics of TN and CTN are as follows:

$$\text{TN} = N_i \times i \text{ and GTN}_j = \sum_{i=1}^{j} (N_i \times i) \tag{4}$$

where *j* is the maximum packet size, and the maximum *j* for Ethernet packets is 1538 bytes.

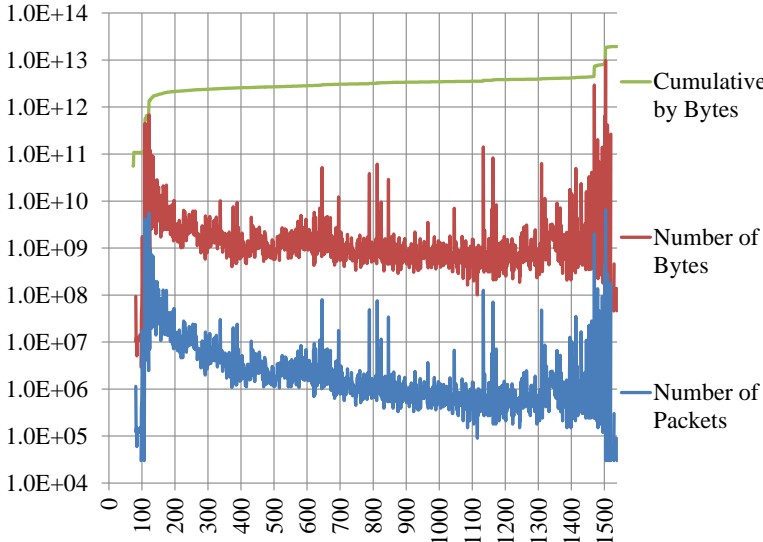

**Figure 12.** Captured traffic of Ethernet packets.

Figure 13 shows the traffic analysis of Ethernet packets. The top of Figure 13 shows the cumulative percentage of packets of various packet sizes, while the bottom shows the cumulative percentage of bytes of various packet sizes. We can see that when the packet size is less than 1300 bytes, the cumulative percentage of bytes is only about 20%, but the cumulative percentage of packets is as high as 68%. The distribution of Ethernet packets of various packet sizes is shown in Figure 14. For example, a packet size group from 100 to 179 bytes is 50.35%, while a packet size group from 1300 to 1538 bytes is 33.21%. This 33.21% of packets has more traffic because these packets are large. The distribution of the traffic model is used to simulate the optical tunnel switching networks. We simulated the performance of the optical tunnel switching networks obtained from a computer. According to the Ethernet data packet distribution shown in Figure 14, the number of data packets transmitted by each server in the optical tunnel switching networks is $10^7$ data packets. In our simulation results, the throughput of an optical tunnel switching network with 25 nodes using one plane and two planes under various loads is shown in Figure 15.

Obviously, the two-plane Torus topology optical switching network is sufficient to provide a throughput close to 100% of the average service throughput of the server.

In our simulation results, the throughput of an optical tunnel switching network with 49 nodes in one plane, two planes and three planes under different loads is shown in Figure 16. Obviously, a three-plane Torus topology optical switching network is sufficient to provide a throughput close to 100% of the average service throughput of the server.

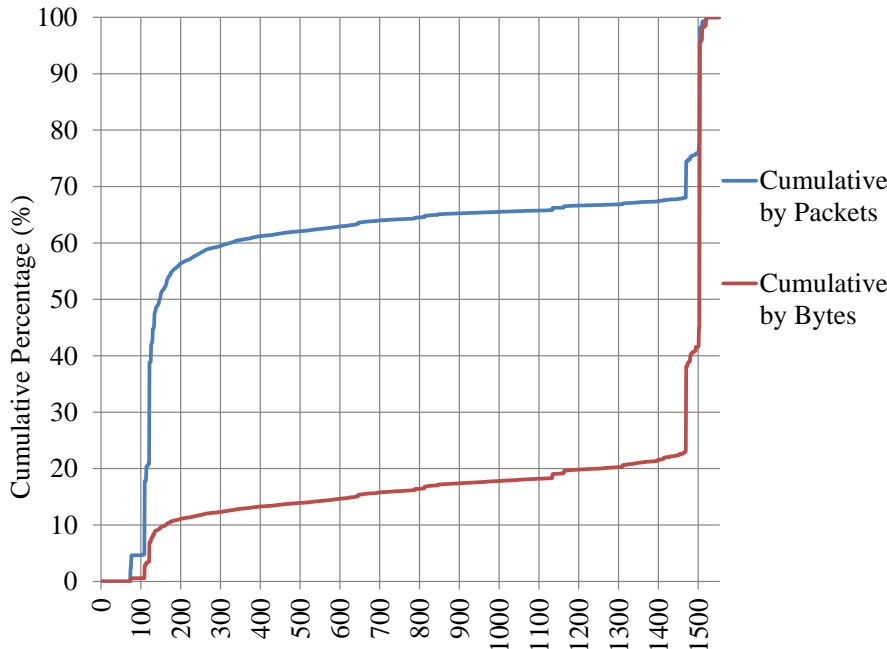

**Figure 13.** Traffic analyses of Ethernet packets.

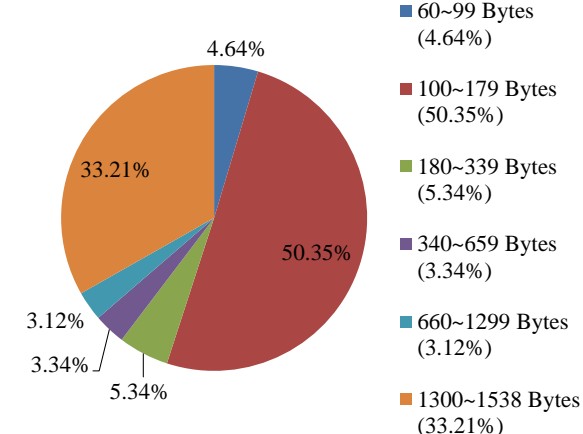

**Figure 14.** Distribution of Ethernet packets.

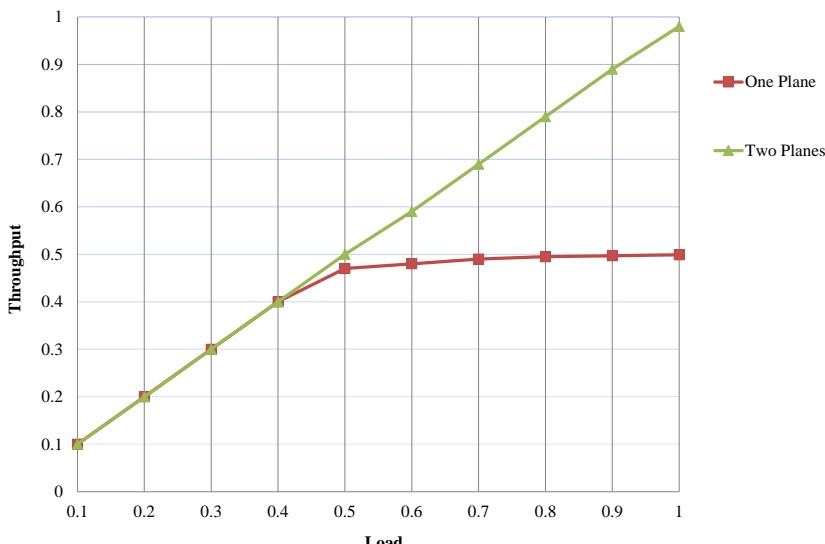

**Figure 15.** Throughput of optical tunnel switching network with 25 nodes.

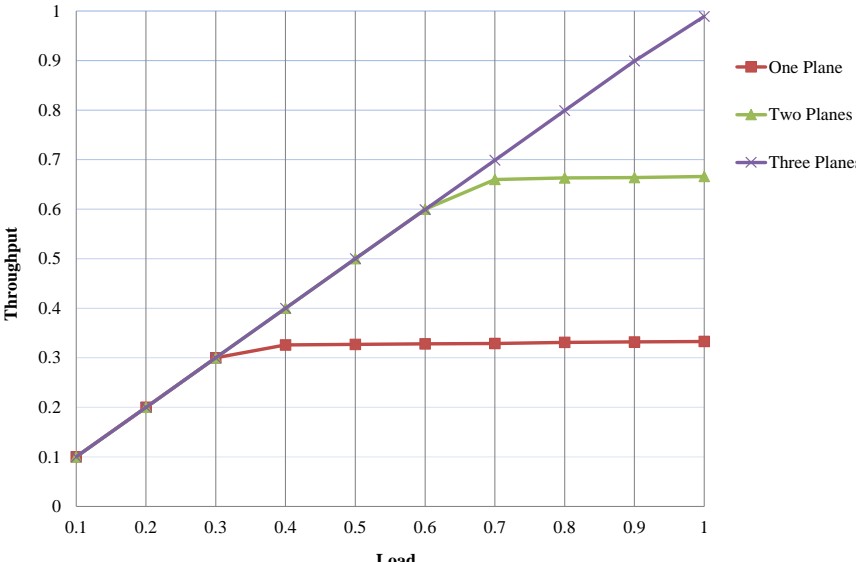

**Figure 16.** Throughput of optical tunnel switching network with 49 nodes.

## 5.2. Traffic Management

Figure 17 shows the traffic distribution of a two-plane optical tunnel switching network with 25 nodes. The maximum available bandwidth between any two nodes is two wavelengths, and the average available bandwidth is 2/3 wavelengths. Generally, the traffic distribution is uneven, so we can combine it with SDN scheduling to meet our bandwidth requirements.

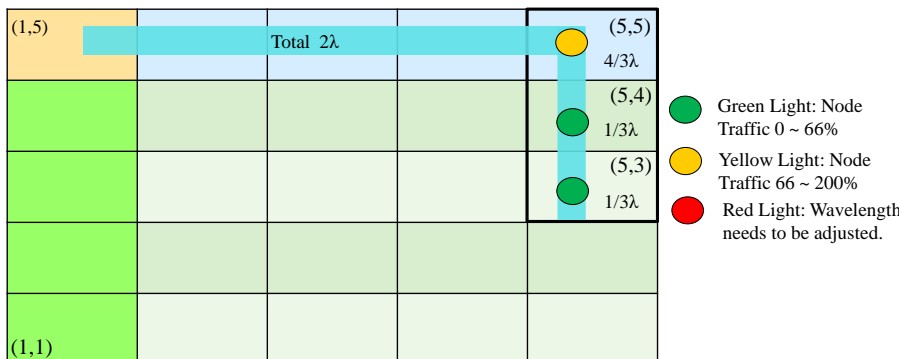

**Figure 17.** Traffic distribution for Method 1of two-plane optical tunnel switching network with 25 nodes.

**Method 1:** We can use three nodes to share the bandwidth of the direct tunnel through SDN scheduling. For example, from node N (1, 5) to node N (5, 5) is 2λ bandwidth, node N (5, 5) uses 4/3 λ bandwidth as well as node N (5, 4), and node N (5, 3) use 1/3 λ bandwidth. This condition can be set on the SDN switch of node N (5, 5).

**Method 2:** We use the wavelengths assigned by neighboring nodes, assuming that the original node goes from node N (1, 5) to node N (2, 5), node N (3, 5), node N (4, 5), and node N (5, 5) uses two wavelengths for communication between nodes. Except for node N (5, 5), which uses 4/3 λ bandwidth, other nodes such as node N (5, 4) and node N (5, 3) use less than 2/3 λ bandwidth, as shown in Figure 18. The details of the change in traffic volume are shown in Figure 19 and described in detail below.

Due to traffic changes, the bandwidth requirement of node N (5, 5) exceeds the 4/3 λ bandwidth. If the sum of the bandwidth of node N (4, 5), node N (4, 2) and node N (4, 1) is equal to or less than the 1λ bandwidth, we can allocate a λ₃ of node N (4, 5) and change it to node N (5, 5), so that a wavelength of 3λ can be used from node N (1, 5) to node N (5, 5), including node N (5, 4) and node N (5, 3), which is shown in Figure 19. However, the total bandwidth used by node N (5, 5) for the entire network is still

limited to 16λ. Therefore, if the bandwidth requirement of a node exceeds the 2λ (200%) bandwidth (red light), including relay tunnel nodes, then we can allocate the neighbor nodes that are smaller than a λ (100%) bandwidth (green light), including relay tunnel nodes to this busy node (red light) to avoid traffic loss.

The traffic management of a three-plane optical tunnel switching network with 49 nodes is similar to the traffic management of a two-plane optical tunnel switching network with 25 nodes.

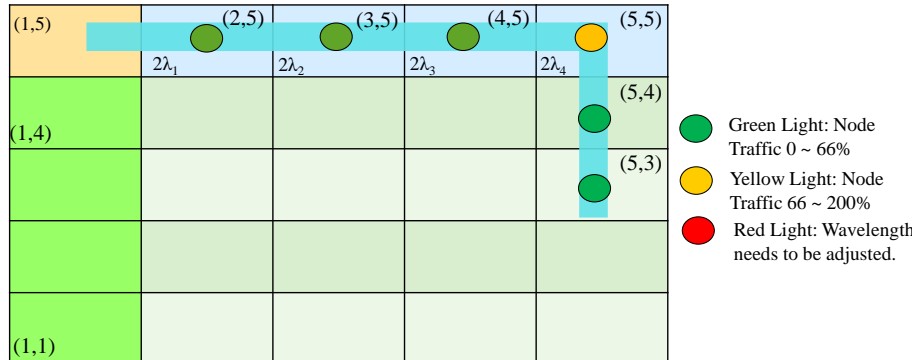

**Figure 18.** Traffic distribution for Method 2 of two-plane optical tunnel switching network with 25 nodes.

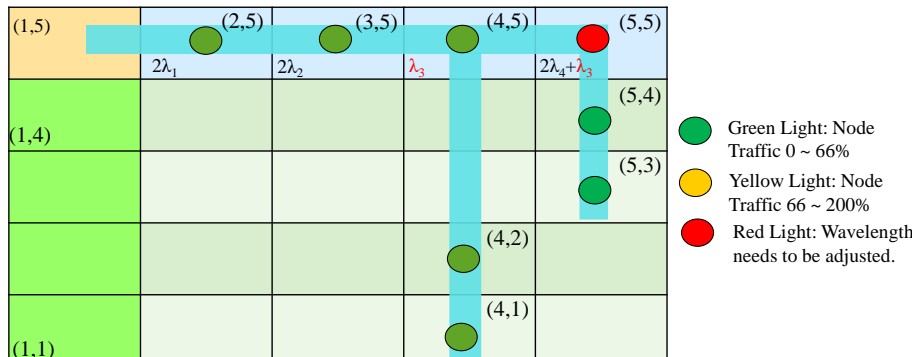

**Figure 19.** Traffic changed of two-plane optical tunnel switching network with 25 nodes.

When the traffic distribution is uneven, in order to handle all operations of traffic changes, the algorithm of real time traffic monitor and management is designed as follows:

---

**Algorithm:** Real Time Traffic Monitor and Management

---

2:       Scanning traffic for all nodes
3: {
4.       If the node's traffic exceeds 200%, then
5:       {
6:             Analyze the traffic of the nodes, and then find the busy tunnel.
7:             If there is a neighboring node whose traffic is less than one λ bandwidth, then
8:             {
9:             Assign a λ bandwidth of the neighboring node to the busy node.
10:             }
11:       }
12: }

---

### 5.3. Applications

Figure 20 shows the application architecture of the Torus topology tunnel switching network in the mobile edge computing (MEC) of Telecom 5G. When high-speed, high-bandwidth, low-latency

applications are required, the traffic is directly transmitted to the server via the Optical Tunnel Switch (OTS), without being sent to the cloud data center for processing of big data. The architecture can provide up to 1764 servers to establish 2352 tunnels. Of course, the aforementioned architecture can also be used in the data center or telecommunications core network to handle big data.

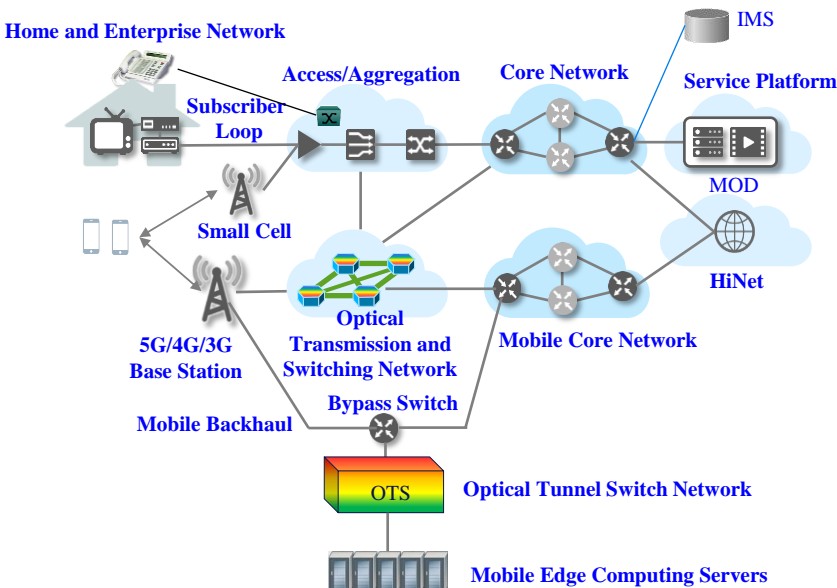

**Figure 20.** Application of Torus topology optical tunnel switching network.

C2TCP (Cellular Controlled Delay TCP) uses TCP to calculate the average delay time with RTT (Round Trip Time) of each packet to control the transmission traffic [13]. Therefore, the multi-plane Torus topology network architecture can also use TCP to calculate the average delay time with RTT of each packet using the delay time of transmitting ACK signals, and then estimate the traffic between nodes, and nodes via the SDN controller, according to the average delay time. When the traffic is changed, the number of network tunnels can be scheduled in time to balance the load traffic to avoid traffic loss.

## 6. Conclusions

A large-scale optical tunnel switching network using Torus topology networks and wavelength selective switches for future big data applications has been presented. All nodes of a large-scale optical tunnel switching network use WSS switching modules, and communication between nodes uses multiple wavelengths to establish a tunnel, and its wavelength can be reused. Because using a wavelength to establish a tunnel not only simplifies the complexity of the network, but also reduces the latency of the network. When the traffic of one node is increased, the number of tunnels for the node can be increased in time to balance the load traffic to avoid traffic loss. Therefore, many applications of the Internet of Things with big data in the future can be realized. The simulation results show that the Torus topology optical switching network with 25 nodes needs two planes to provide approaching to 100% throughput, and the Torus topology optical switching network with 49 nodes needs three planes to provide approaching to 100% throughput. The advantages of the optical tunnel switching network is that it does not require high-speed switching elements and flexible channel scheduling to avoid data loss when traffic volume is changed.

**Author Contributions:** Conceptualization, Y.-J.C., B.-T.C., C.-P.W. and Y.-Y.L.; methodology, Y.-J.C., B.-T.C., and C.-P.W.; software, Y.-J.C., B.-T.C., and C.-P.W.; validation, Y.-J.C., B.-T.C., C.-P.W. and Y.-Y.L.; formal analysis, Y.-J.C., B.-T.C., and C.-P.W.; investigation, Y.-J.C., B.-T.C., and C.-P.W.; resources, Y.-J.C., B.-T.C., C.-P.W. and Y.-Y.L.; data curation, Y.-J.C., B.-T.C., C.-P.W. and Y.-Y.L.; writing—original draft preparation, Y.-J.C., B.-T.C., C.-P.W. and Y.-Y.L.; writing—review and editing, Y.-J.C., B.-T.C., C.-P.W. and Y.-Y.L.; visualization, Y.-J.C.; supervision,

Y.-J.C. and Y.-Y.L.; project administration, Y.-J.C. and Y.-Y.L.; funding acquisition, Telecommunication Laboratories, Chunghwa Telecom Co., Ltd. All authors have read and agreed to the published version of the manuscript.

**Funding:** No external funding was obtained for this research.

**Conflicts of Interest:** There are no conflicts of interest in this research.

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
