# Peer review of "Design of Optical Tunnel Switching Networks for Big Data Applications"

_applsci, doi:10.3390/app10062098_

Round 1

Reviewer 1 Report

This paper deals with a large-scale optical tunnel switching networks based on Torus Topology Network with WSS (Wavelength Selective Switch) for future big data applications. All nodes of the large-scale optical tunnel switching networks use WSS switch modules and the communications between nodes use multiple λs (wavelengths) where a tunnel is established with a wavelength which can be reused. The widely used MEMS and LCoS technologies are all millisecond-level switching speeds, so the frame size of optical frame switch is very large. For example, under a 100 Gbps link, the frame size should be at least 12.5 Mbytes, which will cause a large delay in the optical frame switch, thus affecting the performance of the optical frame switch. Therefore, an optical tunnel switch is proposed. This technology mainly uses DWDM technology to establish many tunnels with multiple wavelengths to provide point-to-point connections. Therefore, optical tunnel switches require many wavelengths to make point-to-point non-blocking connections. This multi-plane Torus topology network architecture not only increases network throughput but also has fault tolerance to increase network reliability. When the traffic is changed, the number of tunnels between nodes can be scheduled in time to balance the load traffic and avoid traffic loss.

However, it would be better to modify following comments before acceptance.

(1) For the use of abbreviations, a full name for the abbreviation must be mentioned once.

(2) There is a duplicated sentence in Abstract and Introduction.
-> “This technology mainly uses DWDM technology to establish many tunnels with multiple wavelengths to provide point-to-point connections. Therefore, optical tunnel switches require many wavelengths to make point-to-point non-blocking connections.”, “For example, under a 100 Gbps link, the frame size should be at least 12.5 Mbytes, which will cause a large delay in the optical frame switch, thus affecting the performance of the optical frame switch. Therefore, an optical tunnel switch is proposed.”

(3) I would like to add a brief description of each chapter of this paper at the end of the introduction.

(4) I wonder why the sentence ‘How many planes are needed depending on the number of nodes and the required throughput?’ is mentioned in chapter 2.

(5) Descriptions of abbreviations such as WSS, MUX, and CDMUX in figure3 should be deleted.

(6) It is necessary to correct a typing error in this paper.
-> To increase the capacity of the network. (‘.’ to ‘,’)

(7) I think that Chapter 5 is close to the contents of Introduction. Authors should consider adding the content of Chapter 5 to Introduction.

(8) Simulation results should be added to clearly show the advantages of the proposed technique. (For example: comparison of performance between Optical Tunnel Switching Network and Optical Frame Switching Network)

Author Response

Response to Reviewer 1 Comments

Point 1: This paper deals with a large-scale optical tunnel switching networks based on Torus Topology Network with WSS (Wavelength Selective Switch) for future big data applications. All nodes of the large-scale optical tunnel switching networks use WSS switch modules and the communications between nodes use multiple λs (wavelengths) where a tunnel is established with a wavelength which can be reused. The widely used MEMS and LCoS technologies are all millisecond-level switching speeds, so the frame size of optical frame switch is very large. For example, under a 100 Gbps link, the frame size should be at least 12.5 Mbytes, which will cause a large delay in the optical frame switch, thus affecting the performance of the optical frame switch. Therefore, an optical tunnel switch is proposed. This technology mainly uses DWDM technology to establish many tunnels with multiple wavelengths to provide point-to-point connections. Therefore, optical tunnel switches require many wavelengths to make point-to-point non-blocking connections. This multi-plane Torus topology network architecture not only increases network throughput but also has fault tolerance to increase network reliability. When the traffic is changed, the number of tunnels between nodes can be scheduled in time to balance the load traffic and avoid traffic loss.

However, it would be better to modify following comments before acceptance.

Point 1:(1) For the use of abbreviations, a full name for the abbreviation must be mentioned once.

Response 1: The full names for the abbreviation must be mentioned once that have been added to the paper (in red) are as follows:

MEMS (Micro-Electro-Mechanical Systems) (line number 15)

LCoS (Liquid Crystal on Silicon) (line number 16)

OPS/OCS (Optical Packet Switch/ Optical Circuit Switch) (line number 49)

SOA (Semiconductor Optical Amplifier) (line number 70)

DWDM (Dense Wavelength Division Multiplexer) (line number 75)

AI (Artificial Intelligence) (line number 101)

SDN (Software Defined Networking) (line number 143)

MUXs (Multiplexer) (line number 199)

CDMUXs (Cyclic De-Multiplexer) (line number 199-200)

Point 2: (2) There is a duplicated sentence in Abstract and Introduction.

-> “This technology mainly uses DWDM technology to establish many tunnels with multiple wavelengths to provide point-to-point connections. Therefore, optical tunnel switches require many wavelengths to make point-to-point non-blocking connections.”, “For example, under a 100 Gbps link, the frame size should be at least 12.5 Mbytes, which will cause a large delay in the optical frame switch, thus affecting the performance of the optical frame switch. Therefore, an optical tunnel switch is proposed.”

Response 2: A duplicated sentence in Abstract has been cancelled and revised some sentence in the paper (in red) as follows: (line number 15-24)

The widely used MEMS (Micro-Electro-Mechanical Systems) and LCoS (Liquid Crystal on Silicon) technologies are all millisecond-level switching speeds, so the frame size of optical frame switch is very large which will reduced switching performance. Therefore, they are only suitable for optical tunnel switching networks design and not suitable for optical frame switch design. For example, under a 100 Gbps link, the frame size should be at least 12.5 Mbytes, which will cause a large delay in the optical frame switch, thus affecting the performance of the optical frame switch. Therefore, an optical tunnel switch is proposed. This technology mainly uses DWDM technology to establish many tunnels with multiple wavelengths to provide point-to-point connections. Therefore, optical tunnel switches require many wavelengths to make point-to-point non-blocking connections.

Point 3: (3) I would like to add a brief description of each chapter of this paper at the end of the introduction.

Response 3: A brief description of each chapter at the end of the introduction has been added to the paper (in red) are as follows: (line number 116-121)

The brief descriptions of other sections are described as follows. In Section 2, we describe operations of multi-plane Torus topology optical tunnel switching network. In Section 3, two-plane Torus topology optical tunnel switching network with 25 nodes is presented and its performance is discussed. The three-plane Torus topology optical tunnel switching network with 49 nodes is presented and its performance is also discussed in Section 4. Finally, the applications of optical tunnel switching network are discussed in Section 5 and some conclusions are made in Section 6.

Point 4: (4) I wonder why the sentence ‘How many planes are needed depending on the number of nodes and the required throughput?’ is mentioned in chapter 2.

Response 4: Revised to “How many planes are needed depending on the number of nodes and the required throughput that will be discussed in Section 3 and Section 4”. (line number 165-166) and described as follows:

If only one plane, the throughput is very low that be described in Section 3 and 4. For example, sharing a direct tunnel between the three nodes reduces the average throughput of traffic to 33% approximately for 25 nodes. Therefore, it is necessary to add a plane Torus topology optical tunnel switching network to increase average throughput of traffic reaching to 100%. For 49 nodes, it is necessary that use three plane Torus topology optical tunnel switching network to increase average throughput of traffic reaching to 100%.

Point 5: (5) Descriptions of abbreviations such as WSS, MUX, and CDMUX in figure3 should be deleted.

Response 5: Descriptions of abbreviations for WSS, MUX, and CDMUX in figure3, figure5, and figure7 have be deleted. (line number 197, 245, and 343)

Point 6: (6) It is necessary to correct a typing error in this paper.

-> To increase the capacity of the network. (‘.’ to ‘,’)

Response 6: “To increase the capacity of the network.” has revised with “To increase the capacity of the network,” that is shown in the paper (in red) (line number 302)

Point 7: (7) I think that Chapter 5 is close to the contents of Introduction. Authors should consider adding the content of Chapter 5 to Introduction.

Response 7: We have added the content of Chapter 5 to Introduction as follows. (line number 79-86) and revised Chapter 5. (line number 443-452)

A large-scale optical tunnel switching network using Torus topology networks and wavelength selective switches for future big data applications is proposed. All nodes of a large-scale optical tunnel switching network use WSS switching modules, and communication between nodes uses multiple wavelengths to establish a tunnel, and its wavelength can be reused. Because using a wavelength to establish a tunnel not only simplifies the complexity of the network but also reduces the latency of the network. When the traffic of one node is increased, the number of tunnels for the node can be increased in time to balance the load traffic to avoid traffic loss. Therefore, many applications of the Internet of Things with big data in the future can be realized.

Point 8: (8) Simulation results should be added to clearly show the advantages of the proposed technique. (For example: comparison of performance between Optical Tunnel Switching Network and Optical Frame Switching Network)

Response 8: We have added some descriptions in Introduction as follows. (line number 87-115)

The proposed architecture of the optical tunnel switching network in this paper is to establish a wavelength tunnel between nodes. This wavelength tunnel is not used by different nodes in time sharing, but the tunnel can be shared by multiple nodes. So it is not necessary to simulate its performance but need to monitor the traffic of each node. When the traffic volume of the node exceeds the traffic volume of the wavelength tunnel, the wavelengths of other nodes with less traffic volume can be scheduled for use. Therefore, we only calculate the traffic volume between nodes and the average available traffic of the server. When the average available traffic is exceeded, wavelength scheduling is performed. For example, if the two plan optical tunnel switching network is adopted, there are two direct tunnels between nodes. The two direct tunnels can be used to communicate between nodes and they are responsibility to establish some relay tunnels for other nodes because tunnels are must be established between all nodes in our proposed architecture. If the optical tunnel switching network has 49 nodes, then 49×48 tunnels are must be established between all nodes. Therefore, if a tunnel traffic is exceeded, we can assign other tunnels with less traffic to support this busy tunnel. The all operations will be handled by a smart scheduler which may be a SDN controller including an algorithm or methodology with AI technology.

The optical frame switching network allows each wavelength channel to be used by different nodes at different times. Therefore, a scheduling algorithm must be used to control the usage of every node and simulate its performance as shown in the paper of reference [13]. Its performance is related to the frame size. Because of the smaller frame with the better performance, therefore, high-speed switching elements such as SOA switching elements are required. The optical tunnel switching network is only used for wavelength tunnel scheduling and does not require high-speed switching elements. Therefore, the performance of the optical tunnel switching network is necessary that average throughput of traffic reaches to 100% at least. However, the performance of the optical frame switching network that average throughput is less than 100% because aggregation algorithm is adopted. Because when much traffic will be sent to the same node, some traffic will be dropped. In the optical tunnel switching network, when much traffic will be sent to the same node, we can schedule wavelength tunnel to avoid traffic loss. The advantages of the optical tunnel switching network is that it does not require high-speed switching elements and flexible channel scheduling to avoid data loss when traffic volume is changed.

Reviewer 2 Report

The paper presents an architecture that seems to have been already described in referenced previous paper. The description of added material with respect to previous work is not enough supported. The presented study should be better motivated in the introduction, that need to be re-shaped mentioning what has been done and what need stil to be done. A mention at paper organization is worth to be added at the end of the introduction.

Instead, a 'Related work' section would help to report about what has been done and previous results.

The remaining part of the paper, although interesting, seems more a report thn a scientific contribution. No methodology neither numerical evaluations are reported.

The references must be fixed (repeated reference numbers).

As a consequence I think that the paper in its present form is not suitable for publication.

Author Response

Response to Reviewer 2 Comments

Point 1: The paper presents an architecture that seems to have been already described in referenced previous paper.

Response 1: Referenced previous paper is that the optical frame switching network allows each wavelength channel to be used by different nodes at different times. Therefore, a scheduling algorithm must be used to control the usage of every node and simulate its performance as shown in the paper of reference [13]. The proposed architecture of the optical tunnel switching network in this paper is to establish a wavelength tunnel between nodes. This wavelength tunnel is not used by different nodes in time sharing, but the tunnel can be shared by multiple nodes. We add some descriptions in the introduction (in red) as follows. (line number 87-115)

The proposed architecture of the optical tunnel switching network in this paper is to establish a wavelength tunnel between nodes. This wavelength tunnel is not used by different nodes in time sharing, but the tunnel can be shared by multiple nodes. So it is not necessary to simulate its performance but need to monitor the traffic of each node. When the traffic volume of the node exceeds the traffic volume of the wavelength tunnel, the wavelengths of other nodes with less traffic volume can be scheduled for use. Therefore, we only calculate the traffic volume between nodes and the average available traffic of the server. When the average available traffic is exceeded, wavelength scheduling is performed. For example, if the two plan optical tunnel switching network is adopted, there are two direct tunnels between nodes. The two direct tunnels can be used to communicate between nodes and they are responsibility to establish some relay tunnels for other nodes because tunnels are must be established between all nodes in our proposed architecture. If the optical tunnel switching network has 49 nodes, then 49×48 tunnels are must be established between all nodes. Therefore, if a tunnel traffic is exceeded, we can assign other tunnels with less traffic to support this busy tunnel. The all operations will be handled by a smart scheduler which may be a SDN controller including an algorithm or methodology with AI technology.

The optical frame switching network allows each wavelength channel to be used by different nodes at different times. Therefore, a scheduling algorithm must be used to control the usage of every node and simulate its performance as shown in the paper of reference [13]. Its performance is related to the frame size. Because of the smaller frame with the better performance, therefore, high-speed switching elements such as SOA switching elements are required. The optical tunnel switching network is only used for wavelength tunnel scheduling and does not require high-speed switching elements. Therefore, the performance of the optical tunnel switching network is necessary that average throughput of traffic reaches to 100% at least. However, the performance of the optical frame switching network that average throughput is less than 100% because aggregation algorithm is adopted. Because when much traffic will be sent to the same node, some traffic will be dropped. In the optical tunnel switching network, when much traffic will be sent to the same node, we can schedule wavelength tunnel to avoid traffic loss. The advantages of the optical tunnel switching network is that it does not require high-speed switching elements and flexible channel scheduling to avoid data loss when traffic volume is changed.

Point 2: The description of added material with respect to previous work is not enough supported.

Response 2: We have added material with respect to previous work in the introduction (in red) as follows. (line number 79-115)

A large-scale optical tunnel switching network using Torus topology networks and wavelength selective switches for future big data applications is proposed. All nodes of a large-scale optical tunnel switching network use WSS switching modules, and communication between nodes uses multiple wavelengths to establish a tunnel, and its wavelength can be reused. Because using a wavelength to establish a tunnel not only simplifies the complexity of the network but also reduces the latency of the network. When the traffic of one node is increased, the number of tunnels for the node can be increased in time to balance the load traffic to avoid traffic loss. Therefore, many applications of the Internet of Things with big data in the future can be realized.

The proposed architecture of the optical tunnel switching network in this paper is to establish a wavelength tunnel between nodes. This wavelength tunnel is not used by different nodes in time sharing, but the tunnel can be shared by multiple nodes. So it is not necessary to simulate its performance but need to monitor the traffic of each node. When the traffic volume of the node exceeds the traffic volume of the wavelength tunnel, the wavelengths of other nodes with less traffic volume can be scheduled for use. Therefore, we only calculate the traffic volume between nodes and the average available traffic of the server. When the average available traffic is exceeded, wavelength scheduling is performed. For example, if the two plan optical tunnel switching network is adopted, there are two direct tunnels between nodes. The two direct tunnels can be used to communicate between nodes and they are responsibility to establish some relay tunnels for other nodes because tunnels are must be established between all nodes in our proposed architecture. If the optical tunnel switching network has 49 nodes, then 49×48 tunnels are must be established between all nodes. Therefore, if a tunnel traffic is exceeded, we can assign other tunnels with less traffic to support this busy tunnel. The all operations will be handled by a smart scheduler which may be a SDN controller including an algorithm or methodology with AI technology.

The optical frame switching network allows each wavelength channel to be used by different nodes at different times. Therefore, a scheduling algorithm must be used to control the usage of every node and simulate its performance as shown in the paper of reference [13]. Its performance is related to the frame size. Because of the smaller frame with the better performance, therefore, high-speed switching elements such as SOA switching elements are required. The optical tunnel switching network is only used for wavelength tunnel scheduling and does not require high-speed switching elements. Therefore, the performance of the optical tunnel switching network is necessary that average throughput of traffic reaches to 100% at least. However, the performance of the optical frame switching network that average throughput is less than 100% because aggregation algorithm is adopted. Because when much traffic will be sent to the same node, some traffic will be dropped. In the optical tunnel switching network, when much traffic will be sent to the same node, we can schedule wavelength tunnel to avoid traffic loss. The advantages of the optical tunnel switching network is that it does not require high-speed switching elements and flexible channel scheduling to avoid data loss when traffic volume is changed.

Point 3: The presented study should be better motivated in the introduction, that need to be re-shaped mentioning what has been done and what need stil to be done.

Response 3: We have added better motivated and re-shaped mentioning what has been done and what need still to be done such as a smart scheduler in the introduction (in red) as follows. (line number 79-115)

A large-scale optical tunnel switching network using Torus topology networks and wavelength selective switches for future big data applications is proposed. All nodes of a large-scale optical tunnel switching network use WSS switching modules, and communication between nodes uses multiple wavelengths to establish a tunnel, and its wavelength can be reused. Because using a wavelength to establish a tunnel not only simplifies the complexity of the network but also reduces the latency of the network. When the traffic of one node is increased, the number of tunnels for the node can be increased in time to balance the load traffic to avoid traffic loss. Therefore, many applications of the Internet of Things with big data in the future can be realized.

The proposed architecture of the optical tunnel switching network in this paper is to establish a wavelength tunnel between nodes. This wavelength tunnel is not used by different nodes in time sharing, but the tunnel can be shared by multiple nodes. So it is not necessary to simulate its performance but need to monitor the traffic of each node. When the traffic volume of the node exceeds the traffic volume of the wavelength tunnel, the wavelengths of other nodes with less traffic volume can be scheduled for use. Therefore, we only calculate the traffic volume between nodes and the average available traffic of the server. When the average available traffic is exceeded, wavelength scheduling is performed. For example, if the two plan optical tunnel switching network is adopted, there are two direct tunnels between nodes. The two direct tunnels can be used to communicate between nodes and they are responsibility to establish some relay tunnels for other nodes because tunnels are must be established between all nodes in our proposed architecture. If the optical tunnel switching network has 49 nodes, then 49×48 tunnels are must be established between all nodes. Therefore, if a tunnel traffic is exceeded, we can assign other tunnels with less traffic to support this busy tunnel. The all operations will be handled by a smart scheduler.

The optical frame switching network allows each wavelength channel to be used by different nodes at different times. Therefore, a scheduling algorithm must be used to control the usage of every node and simulate its performance as shown in the paper of reference [13]. Its performance is related to the frame size. Because of the smaller frame with the better performance, therefore, high-speed switching elements such as SOA switching elements are required. The optical tunnel switching network is only used for wavelength tunnel scheduling and does not require high-speed switching elements. Therefore, the performance of the optical tunnel switching network is necessary that average throughput of traffic reaches to 100% at least. However, the performance of the optical frame switching network that average throughput is less than 100% because aggregation algorithm is adopted. Because when much traffic will be sent to the same node, some traffic will be dropped. In the optical tunnel switching network, when much traffic will be sent to the same node, we can schedule wavelength tunnel to avoid traffic loss. The advantages of the optical tunnel switching network is that it does not require high-speed switching elements and flexible channel scheduling to avoid data loss when traffic volume is changed.

Point 4: A mention at paper organization is worth to be added at the end of the introduction.

Response 4: The paper organization at the end of the introduction has been added to the paper (in red) are as follows: (line number 116-121)

The brief descriptions of other sections are described as follows. In Section 2, we describe operations of multi-plane Torus topology optical tunnel switching network. In Section 3, two-plane Torus topology optical tunnel switching network with 25 nodes is presented and its performance is discussed. The three-plane Torus topology optical tunnel switching network with 49 nodes is presented and its performance is also discussed in Section 4. Finally, the applications of optical tunnel switching network are discussed in Section 5 and some conclusions are made in Section 6

Point 5: Instead, a 'Related work' section would help to report about what has been done and previous results.

Response 5: We have added about what has been done and previous results in the introduction (in red) as follows. (line number 79-115)

A large-scale optical tunnel switching network using Torus topology networks and wavelength selective switches for future big data applications is proposed. All nodes of a large-scale optical tunnel switching network use WSS switching modules, and communication between nodes uses multiple wavelengths to establish a tunnel, and its wavelength can be reused. Because using a wavelength to establish a tunnel not only simplifies the complexity of the network but also reduces the latency of the network. When the traffic of one node is increased, the number of tunnels for the node can be increased in time to balance the load traffic to avoid traffic loss. Therefore, many applications of the Internet of Things with big data in the future can be realized.

The proposed architecture of the optical tunnel switching network in this paper is to establish a wavelength tunnel between nodes. This wavelength tunnel is not used by different nodes in time sharing, but the tunnel can be shared by multiple nodes. So it is not necessary to simulate its performance but need to monitor the traffic of each node. When the traffic volume of the node exceeds the traffic volume of the wavelength tunnel, the wavelengths of other nodes with less traffic volume can be scheduled for use. Therefore, we only calculate the traffic volume between nodes and the average available traffic of the server. When the average available traffic is exceeded, wavelength scheduling is performed. For example, if the two plan optical tunnel switching network is adopted, there are two direct tunnels between nodes. The two direct tunnels can be used to communicate between nodes and they are responsibility to establish some relay tunnels for other nodes because tunnels are must be established between all nodes in our proposed architecture. If the optical tunnel switching network has 49 nodes, then 49×48 tunnels are must be established between all nodes. Therefore, if a tunnel traffic is exceeded, we can assign other tunnels with less traffic to support this busy tunnel. The all operations will be handled by a smart scheduler which may be a SDN controller including an algorithm or methodology with AI technology.

The optical frame switching network allows each wavelength channel to be used by different nodes at different times. Therefore, a scheduling algorithm must be used to control the usage of every node and simulate its performance as shown in the paper of reference [13]. Its performance is related to the frame size. Because of the smaller frame with the better performance, therefore, high-speed switching elements such as SOA switching elements are required. The optical tunnel switching network is only used for wavelength tunnel scheduling and does not require high-speed switching elements. Therefore, the performance of the optical tunnel switching network is necessary that average throughput of traffic reaches to 100% at least. However, the performance of the optical frame switching network that average throughput is less than 100% because aggregation algorithm is adopted. Because when much traffic will be sent to the same node, some traffic will be dropped. In the optical tunnel switching network, when much traffic will be sent to the same node, we can schedule wavelength tunnel to avoid traffic loss. The advantages of the optical tunnel switching network is that it does not require high-speed switching elements and flexible channel scheduling to avoid data loss when traffic volume is changed.

Point 6: The remaining part of the paper, although interesting, seems more a report thn a scientific contribution. No methodology neither numerical evaluations are reported.

Response 6: A smart scheduler which may be a SDN controller including an algorithm or methodology with AI technology will handle entire operations of optical tunnel switching network that be shown as follows and added in the introduction (in red). (line number 87-101).We also calculate the traffic volume between nodes and the average available traffic of the server in Section 3 and 4. If the optical tunnel switching network has 49 nodes, then 49×48 (2,352) tunnels are must be established between all nodes and it is very complexity to handle their operations. If we use three planes with 49×48×3 (7,056) tunnels, the entire operations of optical tunnel switching network are difficulty to assign other tunnels with less traffic to support the busy tunnel that may be need an algorithm or methodology with AI technology.

The proposed architecture of the optical tunnel switching network in this paper is to establish a wavelength tunnel between nodes. This wavelength tunnel is not used by different nodes in time sharing, but the tunnel can be shared by multiple nodes. So it is not necessary to simulate its performance but need to monitor the traffic of each node. When the traffic volume of the node exceeds the traffic volume of the wavelength tunnel, the wavelengths of other nodes with less traffic volume can be scheduled for use. Therefore, we only calculate the traffic volume between nodes and the average available traffic of the server. When the average available traffic is exceeded, wavelength scheduling is performed. For example, if the two plan optical tunnel switching network is adopted, there are two direct tunnels between nodes. The two direct tunnels can be used to communicate between nodes and they are responsibility to establish some relay tunnels for other nodes because tunnels are must be established between all nodes in our proposed architecture. If the optical tunnel switching network has 49 nodes, then 49×48 tunnels are must be established between all nodes. Therefore, if a tunnel traffic is exceeded, we can assign other tunnels with less traffic to support this busy tunnel. The all operations will be handled by a smart scheduler which may be a SDN controller including an algorithm or methodology with AI technology.

Point 7: The references must be fixed (repeated reference numbers).

Response 7: The references have been revised for repeated reference numbers and their formats. (line number 454-497) (line number 46, 48, 51, 65, 368, and 381)

Point 8: As a consequence I think that the paper in its present form is not suitable for publication.

Response 8: The form has been revised as the template of Applied Sciences. We also have added some descriptions in the introduction (in red). (line number 79-115)

Round 2

Reviewer 1 Report

The authors have made a significant improvement in the revised version. I have still some comments.

(1) The excellent performance or advantages of the proposed technique should be demonstrated with simulation results. That is, simulation results should be added.

(2) There are a lot of duplicated sentences in this paper.

(3) I think that the chapter 5 is close to the contents of Introduction. I hope to know the reason that author deals with contents of last 4 paragraphs of chapter 5.

Author Response

The authors have made a significant improvement in the revised version. I have still some comments.

Point 1: (1) The excellent performance or advantages of the proposed technique should be demonstrated with simulation results. That is, simulation results should be added.

Response 1: In Section 5 Performance Evaluation and Traffic Management, the simulation results have been added in the paper (in red) (line number 430-482) (add line number 684-687 for conclusions) (line number 544-557) as well as a traffic management algorithm is designed and added in the paper (line number 544-557) (in red) as follows:

Scanning traffic of all nodes

{

If the traffic of the node is more than 200% then

{

          Analysis traffic of the node and then finding the tunnel that is busy.

           If there is a neighboring node in which the traffic is less than oneλ bandwidth then

           {

             Assign one λ bandwidth of the neighboring node to the busy node.

           }  

}

}

Point 2: (2) There are a lot of duplicated sentences in this paper.

Response 2: The duplicated sentences is deleted such as (line number 72-86), (line number 96-100), (line number 105-118), and (line number 640-644).

Point 3: (3) I think that the chapter 5 is close to the contents of Introduction. I hope to know the reason that author deals with contents of last 4 paragraphs of chapter 5.

Response 3: We have added the content of Chapter 5 to Introduction (line number 36-57) and deleted them (line number 641-675) in chapter 5.

Reviewer 2 Report

The paper still represents just a report. No enough methodology and results are reported. Not suitable for publication.

Author Response

Point 1: The paper still represents just a report. No enough methodology and results are reported. Not suitable for publication.

Response 1: In Section 5 Performance Evaluation and Traffic Management, the simulation results have been added in the paper (in red) (line number 430-482) (add line number 684-687 for conclusions) (line number 544-557) as well as a traffic management algorithm is designed and added in the paper (line number 544-557) (in red) as follows:

Scanning traffic of all nodes

{

If the traffic of the node is more than 200% then

{

          Analysis traffic of the node and then finding the tunnel that is busy.

           If there is a neighboring node in which the traffic is less than oneλ bandwidth then

           {

             Assign one λ bandwidth of the neighboring node to the busy node.

           }  

}

}

Round 3

Reviewer 1 Report

The manuscript is well modified according to reviewer's comments.

I recommend to publish the paper in this journal.

Some acronyms are missing and inappropriately defined. For examples, AI, SDN...

Author Response

The manuscript is well modified according to reviewer's comments.

I recommend to publish the paper in this journal.

Point 1: Some acronyms are missing and inappropriately defined. For examples, AI, SDN...

Response 1: We have added acronyms for AI (line number 36) and moved acronyms of SDN from (line number 176) to (line number 133) as follows.

Currently, the companies focus on seven AI (Artificial Intelligence) applications including chat bots, process advancement and fraud detection, face recognition, market segmentation, sentiment analysis, and call center assistants. The Xiaomi AI speaker has six radio microphones for all-round radio and listening to voice commands. In addition to voice-activated music, listening to audio books, you can also ask for exchange rates, weather, etc., or ask it to calculate math problems. These applications require high speed, high bandwidth, low latency network systems.

The proposed architecture of the optical tunnel switching network in this paper is to establish a wavelength tunnel between nodes. This wavelength tunnel is not used by different nodes in time sharing, but the tunnel can be shared by multiple nodes. So it is not necessary to simulate its performance but need to monitor the traffic of each node. When the traffic volume of the node which are monitored exceeds the traffic volume of the wavelength tunnel, the wavelengths of other nodes with less traffic volume can be scheduled for use. Therefore, we only calculate the traffic volume between nodes and the average available traffic of the server. When the average available traffic is exceeded, wavelength scheduling is performed. For example, if the two plan optical tunnel switching network is adopted, there are two direct tunnels between nodes. The two direct tunnels can be used to communicate between nodes and they are responsibility to establish some relay tunnels for other nodes because tunnels are must be established between all nodes in our proposed architecture. If the optical tunnel switching network has 49 nodes, then 49×48 tunnels are must be established between all nodes. Therefore, if a tunnel traffic is exceeded, we can assign other tunnels with less traffic to support this busy tunnel. The all operations will be handled by a smart scheduler which may be a SDN (Software Defined Networking) controller including an algorithm or methodology with AI (Artificial Intelligence) technology.

Reviewer 2 Report

Please English check section 5

e.g.

detail -> detailed

For investigating -> To investigate

Section 6 is too short. It should be included perhaps as a subsection into another section

Author Response

Please English check section 5

e.g.

Point 1: detail -> detailed

Response 1: We have revised detail into detailed as follows. (line number 421)

For detailed description on the performance of the switch, Figure 9 shows tunnel allocation of two-plane optical tunnel switching network with 25 nodes. Each node of the 5×5 Torus network has 8 direct tunnels and 16 relay tunnels to transfer data to other nodes. Based on the uniform distribution of traffic, on average that one direct tunnel allows two relay tunnels (16/8) to share bandwidth such as node N (5, 5), node N (5, 4), and node N (5, 3 ) share one direct tunnel that from node N (1, 5) to node N (5, 5) as well as node N (1, 4), node N (2, 4), and node N (4, 4) share another direct tunnel that from node N (1, 5) to node N (1, 4). If a plane only uses a relay tunnel to share the bandwidth of a direct tunnel, the network requires dual planes. Therefore, we can connect to all nodes on the network from node N (1, 5) at the same time. The three nodes will share the bandwidth of a direct tunnel, each node will use 2/3 wavelength bandwidth in average for two planes. One SDN switch per node can use 16 wavelength bandwidths for 16 servers (2×8). If these 16 wavelength bandwidths for 16 servers are transmitted to the other 24 node modules on average, the average traffic is also 2/3 wavelength bandwidth (16/24).

Point 2: For investigating -> To investigate

Response 1: We have revised For simulating into To investigate as follows. (line number 464)

To investigate the performance of the optical tunnel switching network, we capture traffic from the Internet core network of Chunghwa Telecom Co., Ltd, Taiwan (the major vendor of telecommunication service). According to the data captured from core network of traditional Ethernet switch, traffic of Ethernet packets is shown in Figure 10, where X-axis represents various packet sizes. The number of packets for various packet sizes is shown in bottom, the total number of bytes (TN) for various packet sizes is shown in the middle, and the cumulative total number of bytes (CTN) for various packet sizes is shown on top in Figure 10. In other words, if the number of packets equals Ni for various packet sizes that let i denoted packet size, then TN and CTN for various packet sizes are characterized as follows:

Point 3: Section 6 is too short. It should be included perhaps as a subsection into another section

Response 3: We have revised Section 6 as a subsection of Section 5 which revised into Network Performance and Its Applications (line number 405). So, Section 5 includes 5.1 Performance Evaluation (line number 407), 5.2 Traffic Management (line number 541), and 5.3. Applications (line number 635). Section 5 is also rearranged (see line number from 405 to 705).
